# Forcing Diffuse Distributions out of Language Models

**Yiming Zhang**[1]* **Avi Schwarzschild**[1]
**Nicholas Carlini**[2] **Zico Kolter**[1,3] **Daphne Ippolito**[1,2]
[1]Carnegie Mellon University
[2]Google DeepMind
[3]Bosch Center for AI

## Abstract

Despite being trained specifically to follow user instructions, today's instruction-tuned language models perform poorly when instructed to produce random outputs. For example, when prompted to pick a number uniformly between one and ten Llama-2-13B-chat disproportionately favors the number five, and when tasked with picking a first name at random, Mistral-7B-Instruct chooses Avery 40 times more often than we would expect based on the U.S. population. When these language models are used for real-world tasks where diversity of outputs is crucial, such as language model assisted dataset construction, their inability to produce diffuse distributions over valid choices is a major hurdle. In this work, we propose a fine-tuning method that encourages language models to output distributions that are diffuse over valid outcomes. The methods we introduce generalize across a variety of tasks and distributions and make large language models practical for synthetic dataset generation with little human intervention.[1]

## 1 Introduction

Consider a Dungeon Master (DM) trying to use a language model based chatbot to assist in managing their Dungeons & Dragons campaign. The DM asks the chatbot to suggest a random name for a character in the story. The first time she asks, it suggests "Anya," and the second time it also suggests "Anya." In fact, almost 40% of the time, the suggested name will be "Anya" even when the language model is deployed with full random sampling. The DM then tries to use the chatbot to roll a twenty-sided die; over 60% of the dice rolls come up as a 14. Frustrated, the DM gives up and brings out their physical dice.

Instruction-tuned language models are extremely bad at producing random outputs when users want them to. Even when prompts are carefully constructed with instructions that encourage randomness, both state-of-the-art open-source and industry language models output very low-entropy distributions over the valid options. Beyond Dungeons & Dragons, there are many practical applications where diversity across valid options is crucial for language model outputs. For example, when language models are used to answer multiple choice or Likert-scale questions, *a priori* each option should be equally likely. When language models are used for synthetic dataset construction, such as for synthetic biographies (Maini et al., 2024; Yuan et al., 2021) or instruction-tuning training sets (Wang et al., 2023), diversity in the generations is essential but arduous to achieve through mere prompt hacking.

In this work, we examine how far off language model generations are from user expectations of randomness and diversity. While our experiments focus on instruction-tuned models such as Llama-2-chat, our findings do generalize to pre-trained language models

---

*Correspondence: Yiming Zhang, `yimingz3@cs.cmu.edu`.
[1]Code and data are available at `https://github.com/y0mingzhang/diffuse-distributions`.

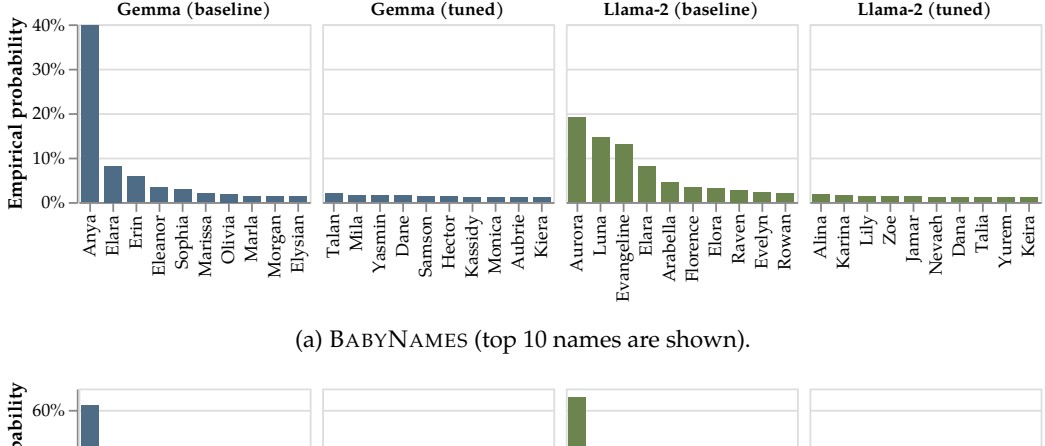

(a) BABYNAMES (top 10 names are shown).

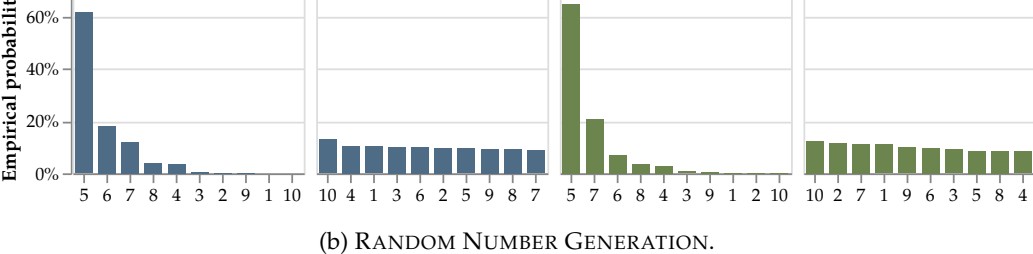

(b) RANDOM NUMBER GENERATION.

Figure 1: **Language models do not produce diffuse probabilities.** The output distributions of untuned Gemma and Llama-2 deviate from what we would expect from natural/random distributions. Our tuning method addresses this by diffusing the output distribution over valid candidates. The horizontal axes above are sorted in descending order by probability of the output.

in general.[2] We then show how language models can be fine-tuned to produce diffuse distributions over valid options, without sacrificing generation quality. Our method supports tasks where the sample set of valid options is not easily enumerated, and we show that models fine-tuned to produce diffuse probabilities for one set of tasks generalize to other, very different tasks without degrading model utility.

This generalization allows us to promote diversity in complex settings, such as synthetic dataset generation. On the task of generating a dataset of synthetic biographies (similar to the dataset created by Maini et al. (2024) for benchmarking language model unlearning), our method generates four times as many unique first names, three times as many unique birth places, and 1.5 times as many unique careers as the untuned baseline model, all without any need for complex prompt engineering, decoding strategy tweaking or manual re-writing.

## 2 Preliminaries

We begin with a formal definition of *diffuse probabilities from language models*. We then introduce techniques for measuring diversity of models outputs and we show how to quantify the differences in the observed output distributions and the desired distributions.

### 2.1 Problem setting

Consider a vocabulary $\mathcal{V} = \{1, 2, ..., n\}$. An autoregressive language model takes a sequence of $\ell$ tokens $\mathbf{x} \in \mathcal{V}^{\ell}$ as input and outputs a probability distribution $\mathbf{p}_{\theta}(\cdot \mid \mathbf{x}) \in \Delta(\mathcal{V})$ over all tokens in the vocabulary. We use $\Delta(\mathcal{V})$ to denote the probability simplex over $\mathcal{V}$.

---

[2]For example, when GPT-2 (Radford et al., 2019) is prompted with "She rolled the 20-sided die. It landed on the number", the output distribution is similarly far from uniform.

In generation, we are interested in computing the probability of a potentially multi-token target $\mathbf{y} = [y_1, y_2, \ldots, y_n] \in \mathcal{V}^n$. For example, $\mathbf{y}$ could be a two-digit number or the biography of a person. A language model factors the probability of a sequence $\mathbf{y}$ into the product of probabilities of individual tokens ($\oplus$ here indicates concatenation):

$$\mathbf{p}_\theta(\mathbf{y} \mid \mathbf{x}) = \mathbf{p}_\theta(y_1 \mid \mathbf{x}) \mathbf{p}_\theta(y_2 \mid \mathbf{x} \oplus y_1) \cdots \mathbf{p}_\theta(y_n \mid \mathbf{x} \oplus \mathbf{y}_{<n})$$

$$= \prod_{i=1}^{n} \mathbf{p}_\theta(y_i \mid \mathbf{x} \oplus \mathbf{y}_{<i}). \tag{1}$$

## 2.2 Measuring output distributions

How do we measure if an LM produces diffuse probabilities? We consider one method for cases where the sample space is enumerable, and another for when the full sample space is unknown.

In settings such as random number generation within a range, we can enumerate the set of valid generations $\mathcal{T}$. For example, in the context of rolling a twenty-sided die, $\mathcal{T} = \{1, 2, \ldots, 20\}$, and we expect a perfectly diffuse language model to produce a uniform distribution over $\mathcal{T}$. More generally, for a context $\mathbf{x}$ and a (not necessarily uniform) ground truth probability distribution $\mathbf{p}^\star \in \Delta(\mathcal{V}^\star)$ over generation targets, we can measure the KL-divergence between the model distribution $\mathbf{p}_\theta$ and the ground truth distribution $\mathbf{p}^\star$ as follows:

$$D_{\mathrm{KL}}\left(\mathbf{p}^\star \parallel \mathbf{p}_\theta\right) = \sum_{\mathbf{y} \in \mathcal{T}} \mathbf{p}^\star(\mathbf{y} \mid \mathbf{x}) \log\left(\frac{\mathbf{p}^\star(\mathbf{y} \mid \mathbf{x})}{\mathbf{p}_\theta(\mathbf{y} \mid \mathbf{x})}\right). \tag{2}$$

KL-divergence measures the difference between the model distribution $\mathbf{p}_\theta$ and the ground truth $\mathbf{p}^\star$, and it attains its minimum at zero if and only if $\mathbf{p}_\theta = \mathbf{p}^\star$.

In many scenarios such as synthetic dataset generation, the support of the ground truth distribution cannot feasibly be enumerated, and we instead measure the *entropy* and the *coverage* of empirical distributions. In these cases, we sample a large number of generations $\mathbf{y}_1, \ldots, \mathbf{y}_N$ from the model and use the empirical distribution $\tilde{\mathbf{p}}$ derived from generations to approximate $\mathbf{p}_\theta$. Entropy of the empirical distribution quantifies the diversity of model generation:

$$\mathrm{Entropy}(\tilde{\mathbf{p}}) = -\sum_{\mathbf{y} \in \mathcal{V}^\star} \tilde{\mathbf{p}}(\mathbf{y}) \log \tilde{\mathbf{p}}(\mathbf{y}). \tag{3}$$

Coverage-$N$ is defined as the number of unique generations produced by the model in $N$ samples:

$$\mathrm{Coverage\text{-}}N(\tilde{\mathbf{p}}) = |\{\mathbf{y}_i \mid i \in [N]\}|. \tag{4}$$

For a sufficiently large $N$, coverage quantifies the number of unique generations a user interacting with the system could practically observe.

# 3 Forcing Diffuse Probabilities

We introduce a method for fine-tuning language models to encourage diffuse probabilities distributions. Our technique hinges on the fact that fine-tuned models generalize—a model which is trained to produce diffuse probabilities for one task will also do so for tasks unseen during training.

## 3.1 Distribution matching as language model fine-tuning

Suppose, for some task, we already know the target set $\mathcal{T}$ of possible outputs, and the ground truth distribution $\mathbf{p}^\star$ we would like the language model to produce. Our method simply maximizes the sum of model log-likelihood over all targets, weighted by their ground truth likelihoods:

$$\mathcal{L}(\mathbf{p}_\theta) = -\sum_{\mathbf{y} \in \mathcal{T}} \mathbf{p}^\star(\mathbf{y} \mid \mathbf{x}) \log \mathbf{p}_\theta(\mathbf{y} \mid \mathbf{x}). \tag{5}$$

This objective is, in fact, the cross-entropy loss ubiquitous in language modeling (Berger et al., 1996) extended to multiple target sequences (Edunov et al., 2018). Since the model distribution $\mathbf{p}_\theta$ is defined on sequences, this optimization relies on a mild condition that $\mathcal{T}$ is prefix-free, namely no target sequence $\mathbf{t}$ should be the prefix of another target sequence $\mathbf{t}'$. This condition is necessary since we would otherwise always have $\mathbf{p}_\theta(\mathbf{t}' \,|\, \mathbf{x}) < \mathbf{p}_\theta(\mathbf{t} \,|\, \mathbf{x})$, and it would be impossible to, say, make a model output identical probability for $\mathbf{t}$ and $\mathbf{t}'$.[3] Notice that minimizing the objective in Eq. 5 is equivalent to minimizing the KL-divergence between the model and the ground truth distribution $\mathbf{p}^\star$:

$$
\begin{aligned}
\mathcal{L}(\mathbf{p}_\theta) &= \sum_{\mathbf{y}\in\mathcal{T}} \mathbf{p}^\star(\mathbf{y} \,|\, \mathbf{x}) \left( \log \frac{\mathbf{p}^\star(\mathbf{y} \,|\, \mathbf{x})}{\mathbf{p}_\theta(\mathbf{y} \,|\, \mathbf{x})} - \log \mathbf{p}^\star(\mathbf{y} \,|\, \mathbf{x}) \right) \\
&= \sum_{\mathbf{y}\in\mathcal{T}} \mathbf{p}^\star(\mathbf{y} \,|\, \mathbf{x}) \log \left( \frac{\mathbf{p}^\star(\mathbf{y} \,|\, \mathbf{x})}{\mathbf{p}_\theta(\mathbf{y} \,|\, \mathbf{x})} \right) - \sum_{\mathbf{y}\in\mathcal{T}} \mathbf{p}^\star(\mathbf{y} \,|\, \mathbf{x}) \log \mathbf{p}^\star(\mathbf{y} \,|\, \mathbf{x}) \\
&= D_{\mathrm{KL}} \left( \mathbf{p}^\star \,\|\, \mathbf{p}_\theta \right) - \mathrm{Entropy}(\mathbf{p}^\star),
\end{aligned}
$$

where the entropy term is independent from the model distribution. In this view, our method can be seen as *distribution matching*, where we align the model distribution with an ideal distribution.

In the more complex case where the ground truth distribution is not known, our method assumes access to a small sample set ($\sim$200 samples) and treats the empirical distribution as a proxy of ground truth. While minimizing this loss to 0 could lead to a model that exclusively produces targets from the sample set, which is not what we want in a case like synthetic dataset generation, empirically, we find this not to be the case. We discuss results on generalization beyond training distribution extensively in Section 5.

## 3.2 Parameter-efficient Fine-tuning

We use LoRA (Hu et al., 2021) to fine-tune models in a parameter-efficient way. This technique relies on optimizing low-rank additions to weight matrices, making fine-tuning large language models (with $\geq$ 7B parameters) practical. It has the added benefit that LoRA tuning tends to perform almost as well as full model fine-tuning and generalizes beyond training instances (Malladi et al., 2023). From an implementational standpoint, LoRA requires instantiating a new pair of relatively small matrices for every weight matrix in the network. Consider one fully connected layer parameterized by $W \in \mathbb{R}^{d\times d}$. We instantiate $B \in \mathbb{R}^{d\times r}$ and $A \in \mathbb{R}^{r\times d}$ where $r \ll d$ and reparametrize the layer as $W + BA$. When fine-tuning the model, we exclusively optimize the weights of $A$ and $B$ rather than taking gradient steps on $W$, enabling us to fine-tune large models such as Llama-2-13B with only a small fraction of the memory overhead.[4] We report training details in Appendix A.

## 4 Results on Simple Tasks

We begin by showing how state-of-the-art language models fail to produce diffuse distributions on simple tasks, even when a user's prompt requests randomness. We consider two tasks: random baby name generation (BABYNAMES) and random number generation (RNG). In BABYNAMES, the model is instructed to generate a random baby name in English. In RNG, we ask the model to generate a random number between 1 and 10. The tasks have the following prompts:

1. BABYNAMES: "Please generate an English first name, chosen completely at random. Output only the name between two curly braces, like this: {name}. Don't output code."

---

[3]We can simply append an end-of-sequence token to each target if $\mathcal{T}$ is not already prefix-free.

[4]Empirically, we find that $r = 4$ is sufficiently expressive, which we use across all experiments. On our models, this corresponds to tuning less than 0.1% of all model parameters.

Table 1: Our method significantly increases the diversity of generations on the random number and baby name generation tasks. Coverage-1000 is the number of unique outcomes out of 1,000 trials.

| Task | Model | Entropy ↑ | | | Coverage-1000 ↑ | | |
|---|---|---|---|---|---|---|---|
| | | Baseline | Tuned | Natural | Baseline | Tuned | Natural |
| RNG | Gemma | 1.16 | 2.28 | | 8 | 10 | |
| | Llama-2 | 1.07 | 2.29 | 2.30 | 7 | 10 | 10 |
| | Mistral | 1.88 | 2.30 | | 9 | 10 | |
| BABYNAMES | Gemma | 3.19 | 5.69 | | 170 | 442 | |
| | Llama-2 | 3.24 | 5.83 | 6.43 | 129 | 490 | 714 |
| | Mistral | 4.02 | 5.78 | | 225 | 479 | |

2. RNG: "Generate a random number between 1 and 10. Output only the number between two curly braces, like this: {number}. Don't output code."[5]

We consider three state-of-the-art language models: Gemma-7B-instruct (Google, 2024), Llama-2-13B-chat (Touvron et al., 2023) and Mistral-7B-instruct (Jiang et al., 2023). We use chat/instruct versions of the models, which are fine-tuned on dialog data to make them more suitable than base models for our test cases, which require instruction following. In all experiments, we sample from models using a temperature of one, which is equivalent to sampling from the maximally diffuse, unmodified model distributions.[6] For each task, we sample 1,000 generations from the model and report coverage and entropy statistics of the empirical distributions of model generations.

## 4.1 Language models fail to follow instructions for randomness

Figure 1 shows output distributions of several untuned, baseline models for BABYNAMES and RNG. Despite the prompts asking for randomness, these output distributions are extremely imbalanced.

For RNG, it is easy to see that the empirical distribution is far from the uniform distribution we would expect. For BABYNAMES, it is less clear what we should expect from the model. Do we want the generated names to be a uniform sample from the set of all names? Or do we want the names to reflect the distribution of baby names among recently born babies in the Unites States? To address this, we introduce the concept of a *natural* distribution for each task. For RNG, this is simply the uniform distribution; for BABYNAMES, we compare to a sample we draw from data on names used for babies born in the United States in the year 2022, using birth data from the US Social Security Administration.[7] We note, that we are only using the natural distributions as reasonable baselines for comparison.

When generating names, we find that Llama-2 generates the name "Aurora" about 100 times more often than we would expect from the natural distribution, and Gemma assigns 40% probability to the name "Anya", which did not appear even once in our sample of the natural distribution. This is a clear indication that these large language models do not sample diversely from a large space of valid generations (all English names) despite being explicitly instructed to do so.

In the RNG task, we find that each model has a bias toward a different set of numbers. For example, both Gemma and Llama-2 assign over 60% probability to the number five, while Mistral assigned over 25% probability to the number eight. Surprisingly, in 1000 samples,

---

[5]We instruct the models to generate between curly braces to simplify output parsing. Gemma has a strong tendency to output code and we add an explicit instruction to avoid code generation.

[6]We could technically set temperature to beyond one to smooth out the model distribution further, but this is rarely done in practice due to degeneration. We also note that the untuned, baseline models need to be sampled at a temperature $> 10$ to match the entropy of our fine-tuned models in Section 4.2.

[7]https://www.ssa.gov/OACT/babynames

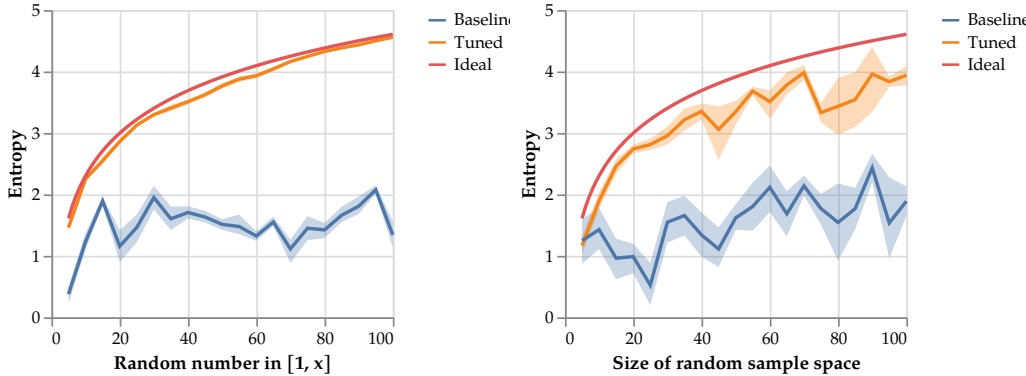

(a) Varying prompt formats and number ranges.    (b) Varying sizes of random sample space.

Figure 2: Models tuned on RANDOM NUMBER GENERATION demonstrate generalization to variations in both prompt format and number ranges. 95% confidence intervals are shown in plots.

none of the models were able to generate all ten numbers. These results highlight that the output distributions of large language models are far from diffuse, even in simple tasks where a user may expect them to be.

## 4.2 Optimizing for BABYNAMES and RNG diversifies generation

We start by focusing on the setting where we are optimizing the model to produce diffuse probabilities for the two individual tasks. For RNG, we fine-tune the model to produce a uniform distribution from one to ten, and for BABYNAMES, we fine-tune the model to fit the empirical distribution of a sample of 200 baby names drawn from the natural distribution.

In these experiments, we see our methods can very effectively align the model distribution with the ideal distributions (Table 1). With RNG, tuned models produce near-uniform distributions over the valid choices with full coverage and near perfect entropy. With BABY-NAMES, where the training set covers only a small portion of the possible outputs, we see an encouraging indication of generalization: while the coverage remains smaller than that of the natural distribution, it is over twice the baseline for all models and the majority of outputs are not among targets in the fine-tuning set.

While RNG and BABYNAMES serve as toy problems, these settings have a major limitation—the fine-tuned models are evaluated on the same prompts and sample spaces in training. To further test the generalization performance of our models, we also consider settings where the prompt and sample space differ between training and evaluation. We fine-tune Llama-2 to produce diffuse probabilities over two ranges of random integers—one to ten and one to 100—and test its generalization to different instruction formats and unseen sample spaces (see Appendix B.1 for prompts). In Figure 2a, we observe encouraging generalization trends: the tuned model produces near uniform distributions for unseen prompt formats over number ranges not in the fine-tuning set, for example from 1 to 45. When we vary the size of the random sample space of RNG (e.g., 154 to 204), the tuned model still produces substantially higher entropy distributions than the untuned model (Figure 2b). With these first experiments, we demonstrate that tuning models for diffuse probabilities is a promising method for increasing generation diversity.

## 5 Generalization Across Tasks

It is not too surprising that a model optimized to output random numbers uniformly across some range can generalize to other ranges of numbers. A much more interesting and practical test is if a model optimized for diffuse probabilities on one set of tasks can transfer to

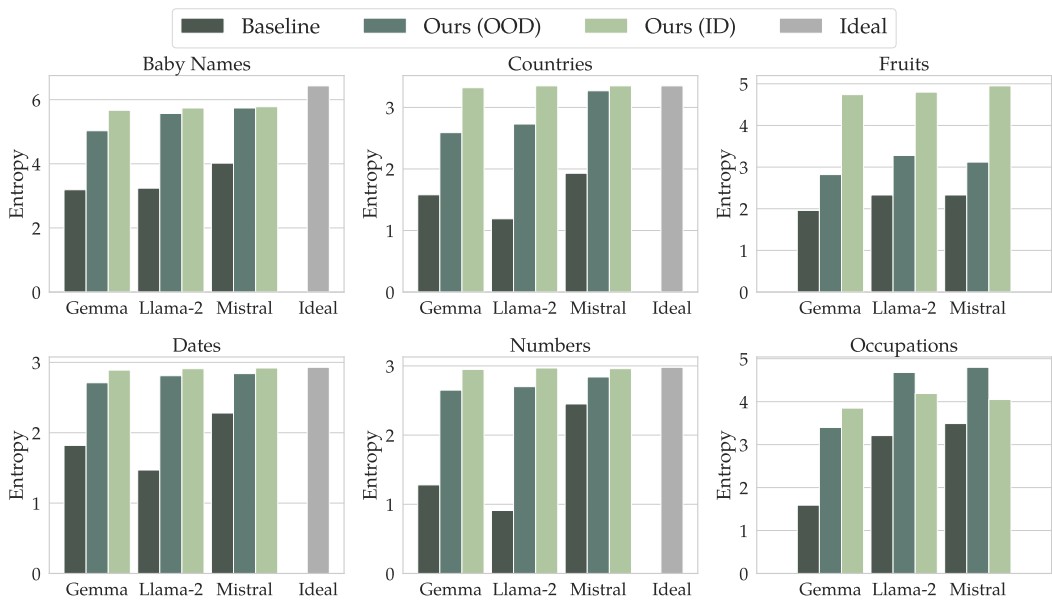

Figure 3: **Entropy in leave-one-out generalization.** The title of each plot indicates which set of tasks we compute entropy over.

tasks with very different sample spaces. That is, a model optimized for picking a random number or baby names should not overfit to generating samples from those distributions; rather, it should be able to e.g. pick a random fruit or country name when prompted to do so.

We use leave-one-out experiments on a set of six tasks to show that models tuned for diffuse probabilities do, in fact, have strong transferability to tasks unseen during tuning. We consider the following six tasks. Each task has several associated prompts, one of which is shown below:

1. BABYNAMES: "Please generate an English first name, chosen completely at random."
2. COUNTRIES: "Output a random country in Africa, chosen completely at random."
3. FRUITS: "Output a name of a fruit, chosen completely at random."
4. DAYS AND DATES: "Provide a random date in June."
5. NUMBERS: "Randomly pick a prime number between 1 and 50."
6. OCCUPATIONS: "Output an occupation that starts with the letter "A"."

By fine-tuning on five out of the six tasks listed above and evaluating performance on the sixth, we can measure the ability of our method to handle out-of-distribution tasks.

In Figure 3, we report in-distribution (ID) and out-of-distribution (OOD) results, which correspond to tuning sets that include or exclude the particular task, respectively. The results show a convincing trend: for all three models, our tuning method led to substantial improvements in entropy over the baselines, even when the task was held out from the tuning set.[8] Another interesting observation is that the baseline Mistral model consistently produces more diffuse distributions than the baseline Gemma and Llama-2 models, but after tuning, all three models have comparable entropy.

In two of the tasks (COUNTRIES and FRUITS), we observe sizable generalization gaps between in-distribution and out-of-distribution entropy, which suggest that task-specific tuning remains useful, especially when we can come up with a reasonbly diverse set of generation targets for fine-tuning. However, coming up with a large enough target set isn't always easy. In these cases, we rely on the generalization of the models trained on a di-

---

[8]Coverage results show similar trends, and we report them in Table 5, Appendix C.1.

Figure 4: **Fine-tuned Llama-2 model improves the diversity of synthetic biographies.** We report coverage for categorical attributes in 4a and normalized unigram diversity of generated achievements and the entire biography in 4b.

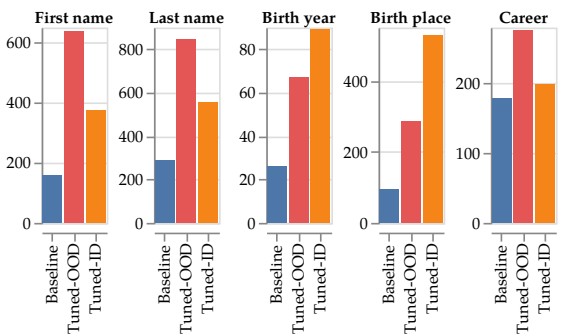

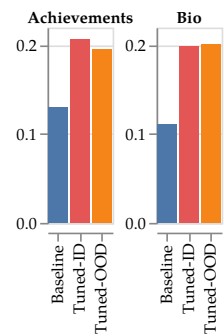

(a) Coverage results for categorical attributes.  (b) Unigram diversity of biographies.

verse set of tasks. For example, in OCCUPATIONS, the authors could only come up with a small set of 17 professions that start with the letter "A," and Llama-2 and Mistral (not trained on OCCUPATIONS) generalize out-of-distribution to occupations beyond what we provide in the fine-tuning set.[9] Notably, our fine-tuning method does not substantially change the general capabilities (e.g., writing and reasoning) of the models demonstrated by evaluations on MT-Bench (see Appendix C.4), making our method compatible with tuning general-purpose large language models.

## 6 Constructing More Diverse Synthetic Datasets

The leave-one-out experiments show that fine-tuning for diffuse outputs leads to task generalization. This is an important trait for real-world applications such as synthetic dataset construction, where it might be not be feasible to tune on data that is identically formatted to what we would like to synthesize.

In this section, we evaluate how well a model tuned on the six tasks from Section 5 performs on a realistic dataset creation task—building a synthetic dataset of fictional biographies. Inspired by the synthetic datasets created in Maini et al. (2024) and Yuan et al. (2021), these biographies include the following attributes: first and last name, gender, birth year, birth place, profession and a description of the person's achievements. We show how our method results in biographical details that are much more diverse than those generated by the baseline models.

### 6.1 Out-of-distribution training improves synthetic dataset diversity

We first consider a Llama-2-13B model fine-tuned on all tasks in Section 5 and not specifically for biographies (Tuned-OOD), and compare its generations against the baseline Llama-2 model over 1000 samples. In Figure 4, we report coverage results for categorical attributes (e.g., Birth year), and normalized unigram diversity for achievements and over the entire biography.[10] In Table 2, we report the most frequently generated values for categorical attributes, along with the frequencies out of 1000 generations.[11]

The results on the baseline Llama-2 model indicate that the biases we observe in Section 4.1 towards certain names and numbers expectedly show up in generation. For example, out

---

[9]For example, Llama-2 generated "Aromatherapist", and Mistral generated "Agronomist." Neither is among the fine-tuning targets of OCCUPATIONS.

[10]We report prompt used for biography generation and side-by-side qualitative examples in Appendix B

[11]A table of coverage and entropy statistics of generated biographies can be found in Appendix C.2.

Table 2: The most frequently generated values for each attribute, along with the number of times the value was generated (out of 1000 generations). Despite being tuned on out-of-domain tasks, the tuned Llama model significantly improves in diversity. *Some values in table have been truncated for brevity.*

| First name | | Last name | | Gender | | Birth year | | Birth place | | Career | |
|---|---|---|---|---|---|---|---|---|---|---|---|
| | | | | | | Baseline Llama-2 | | | | | |
| Evelyn | 284 | Nightingale | 117 | F | 966 | 1985 | 764 | Paris, FR | 305 | Astronaut | 211 |
| Luna | 155 | Aurora | 104 | NB | 17 | 1987 | 46 | Tokyo, JP | 267 | Astro. Engineer | 66 |
| Elara | 87 | Nova | 98 | M | 13 | 1992 | 44 | Stockholm, SE | 33 | Aero. Engineer | 55 |
| Adriana | 42 | Starling | 53 | | | 1978 | 36 | Mumbai, IN | 32 | Env. Activist | 42 |
| Aurora | 38 | Stardust | 41 | | | 1975 | 31 | Singapore, SG | 32 | Astrophysicist | 32 |
| | | | | | | Fine-tuned Llama-2 (OOD) | | | | | |
| Luna | 32 | Nightingale | 16 | F | 762 | 1985 | 211 | Mumbai, IN | 35 | Astronaut | 96 |
| Zelda | 14 | Nightshade | 12 | M | 189 | 1992 | 99 | Lagos, NG | 31 | Aero. Engineer | 50 |
| Mila | 14 | Chen | 8 | NB | 31 | 1987 | 77 | Paris, FR | 29 | Soft. Engineer | 47 |
| Evelyn | 11 | Orion | 6 | | | 1988 | 61 | Tokyo, JP | 27 | Env. Activist | 35 |
| Althea | 9 | Sparks | 6 | | | 1990 | 52 | Nairobi, KE | 21 | Journalist | 34 |
| | | | | | | Fine-tuned Llama-2 (ID) | | | | | |
| Hava | 14 | Kim | 17 | F | 487 | 1921 | 39 | Choloma, HN | 16 | Architect | 140 |
| Maria | 14 | Mohammed | 16 | M | 478 | 1931 | 36 | Rabat, MA | 13 | Journalist | 74 |
| Juan | 14 | Khan | 13 | NB | 34 | 1942 | 35 | Tainan, TW | 10 | Politician | 35 |
| Valcin | 13 | Abed | 13 | | | 1916 | 30 | Rajshahi, BD | 10 | Archaeologist | 35 |
| Issaka | 12 | Salah | 12 | | | 1984 | 29 | Budapest, HU | 10 | Mar. Biologist | 26 |

of 1000 biographies generated by the baseline Llama-2 model, 284 have the first name 'Evelyn,' and 966 are females. Such a high level of repetition makes the resulting dataset basically unusable without substantial human intervention. In contrast, our model significantly improves the diversity of generated biographies, despite not trained specifically for generating biographies. For example, we see an >2X increase in coverage for most categorical attributes. There are also substantial improvements in generation diversity for achievements and over the entire biography (Figure 4b), although our training does not optimize for open-ended text generation.

The top five most frequent values for attributes (Table 2) help contextualize these improvements in diversity: there are significant reductions in biases towards certain attribute values. E.g., the frequency of the name "Evelyn" decreased by 25X, and the birth year 1985 by over 3X.

## 6.2 Diversifying distributions of categorical attributes

Despite this improvement over base model, certain biases, as seen in the high frequencies of female biographies (76.2%) and the birth year 1985 (21.1%), still persist, which could limit the utility of the dataset. Our fine-tuning method can in fact be directly applied to balance the distribution of categorical attributes. As a proof of concept, we create a target set containing 210 programmatically generated tuples of categorical attributes, with roughly balanced gender, birth year and birth place distributions without the open-ended achievement descriptions. We then fine-tune a Llama-2 model (Tuned-ID) only on categorical attributes and evaluate another sample of 1000 biographies.

At a comparable level of generation diversity to the Tuned-OOD model (Figure 4), the Tuned-ID model is able to generate biographies with much more balanced distributions of categorical attributes: Table 2 shows that both gender and birth years are roughly uniformly distributed, and a wider range of birth places are produced by the model.[12] Crucially, the model remains highly diverse on open-ended generation of achievements even when being trained exclusively on categorical attributes. This result highlights the potential in extending our fine-tuning method towards improving diversity in open-ended text generation.

---

[12]See maps of generated birth places for all three models in Figure 5, Appendix C.

# 7 Related Work

**Diversity in Text Generation**   The lack of diversity has been a long-standing issue in generation due to the tension between generation quality and diversity (Zhang et al., 2020): sampling at low temperature causes boring and repetitive text, while sampling at higher temperatures could lead to nonsensical output (Tevet & Berant, 2021). Much of the existing literature approaches the quality-diversity tradeoff by coming up with new decoding strategies, which often involve either shaping the model distribution (e.g., top-$p$ sampling (Holtzman et al., 2020) and top-$k$ sampling (Fan et al., 2018)) and diversity-promoting constraints during decoding (Li et al., 2016; Vijayakumar et al., 2018) or training (Welleck et al., 2019; Edunov et al., 2018). Our setting is distinct from prior work in that we already assume a "maximally" diverse decoding strategy (i.e., sampling from the model with temperature one), and yet instruction-tuned language models still fail to generate diversely.

**Language Model for Dataset Creation**   As more capable language models emerge (Touvron et al., 2023; Jiang et al., 2023; Google, 2024), language model-based dataset creation becomes increasingly practical. Prior work has largely focused on generating specialized data for augmenting NLP tasks (Ye et al., 2022b) including semantic similarity (Schick & Schütze, 2021), relationship extraction (Chia et al., 2022), natural language understanding (Meng et al., 2022) and instruction following (Honovich et al., 2022). A LM-based dataset creation pipeline is usually an iterative and arduous process (Ye et al., 2022a), in which significant human intervention is needed to ensure the model generates diverse and high-quality data (Yuan et al., 2021; Liu et al., 2022; Maini et al., 2024). Our work aims to make progress towards the goal of automating data creation by improving the diversity of language model generation and thereby reducing the need for human intervention.

# 8 Concluding Discussion and Future Directions

In this work, we propose a method for fine-tuning language models to generate diffuse probability distributions, and show that this method leads to sizable and transferrable improvements in generation diversity. We showcase a practical application of our method in synthetic dataset generation, demonstrating improvements in the quality of generated data by a large margin, with or without task-specific tuning. Our experiments reveal interesting insights on the surprising capability of language models to learn diffuse distributions and generalize to new prompts and output spaces. An important direction for future work is the application of our method (and distribution matching techniques in general) to debiasing language models (Liang et al., 2021), which are shown to be rife with harmful stereotypes (Bolukbasi et al., 2016). Given the strong generalization properties of our method, it is plausible that aligning language models with an ideal distribution of representative instances could reduce model bias.

Since our method requires access to a set of valid output sequences, a structured output space is assumed. In other words, this method is not directly applicable to open-ended generation tasks where the number of valid targets is practically infinite. Future work should further explore the limits of our approach, especially in the context of improving diversity of open-ended generation. Although our method is independent from the fine-tuning procedure, we find LoRA (Hu et al., 2021) to be substantially more efficient and effective compared to other techniques such as prefix-tuning (Li & Liang, 2021) and discrete prompt search (Zou et al., 2023). Since they could potentially guide closed-source models to generate more diverse outputs, identifying diversity-inducing discrete prompts is a particularly interesting research question, which we leave for future work.

**Acknowledgments**

We are grateful to the Center For AI Safety for providing compute for experiments in this paper.

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

# A    Training details

For all experiments, we use LoRA to fine-tune models for up to 50 steps over all target prompts. In tasks where the ground truth distribution cannot be enumerated (e.g., synthetic biographies), we stop fine-tuning early when the training loss is within 20% of the minimum loss, which is the entropy of the target distribution. We use an AdamW optimizer (Loshchilov & Hutter, 2019) with a learning rate of $5 \times 10^{-5}$ and a batch size of 32. All experiments are conducted on a single NVIDIA A100 GPU, using a total of approximately 50 GPU hours. Code and prompts used to reproduce our results are available at the open-source GitHub repository.[13]

# B    Prompts and Qualitative Results

## B.1    Prompts used for Figures 2

The y-axis in Figure 2a gives the average entropy over the following five prompts, where low is set to 1, and high is the value on the x-axis.

1. "Randomly pick an integer between low and high. Do not provide any code. Output only the number between two curly braces, like this: {number}. Don't output code."
2. "Output a number uniform randomly from the range low to high, inclusive. Output the number between two curly braces, like this: {number}. Don't output code."
3. "Choose a number at random between low and high. Output your chosen number bweet two curly braces, like this {number}."
4. "Select an integer completely at random from the interval [low, high]. Output the selected integer between two curly braces, like this {n}."
5. "I need you to pick a number completely at random. The number should be between low and high. Print the number between two curly braces, like this: {n}."

The y-axis in Figure 2b gives the average entropy over five versions of Prompt 1 above, where low is chosen randomly to be between 1 and 900, and high is the value on the x-axis.

---

[13]https://github.com/y0mingzhang/diffuse-distributions

## B.2 Prompt used for synthetic biography generation

Table 3: Prompt used for synthetic biography generation.

---

Generate a random biography sketch of a fictional, notable person. Output name, gender, time of birth, place of birth, profession and accomplishments individually between two braces and generate nothing else. Please follow the format below.

{First name and last name}
{Gender - Female/Male/Non-binary}
{Date and year of birth}
{City and country of birth}
{Profession}
{One sentence description of accomplishments}

Example
{David Hippocampus}
{Male}
{June 30, 1960}
{Cambridge, Massachusetts, USA}
{Neuroscientist}
{Revolutionized cognitive science by elucidating the mechanisms underlying memory formation and retrieval through groundbreaking research and innovative methodologies, advancing our understanding of the human brain's intricacies}

---

## B.3    Side-by-side comparison of generated biographies across systems

In Table 4, we show five randomly sampled biographies generated by the baseline Llama-2 model and our fine-tuned models, both out-of-distribution and in-distribution. We note that just in 5 examples, there are noticeable quality issues with the baseline model, such as repetition of the same birth date, birth place and professions, as well as largely similar descriptions of accomplishments. In contrast, our fine-tuned models generate more diverse biographies.

Table 4: Five randomly sampled biographies generated by the baseline Llama-2 model and our fine-tuned models.

| Llama-2 (Baseline) | Llama-2 (Tuned-OOD) | Llama-2 (Tuned-ID) |
|---|---|---|
| {Cesar Season}
{Male}
{March 22, 1985}
{Montevideo, Uruguay}
{Chef}
{"Reimagined the flavors of South American cuisine with his avant-garde dishes, earning him numerous accolades and characteristics for his culinary innovation and experimentation."} | {Teagan Sunshine}
{Female}
{July 12, 1998}
{Copenhagen, Denmark}
{Chef}
{Pioneered a sustainable, seasonal, and socially conscious gastronomy movement by founding a Michelin-starred restaurant that sources exclusively from local farmers} | {Flick Monteiro}
{Female}
{March 18, 1942}
{Vaduz, Liechtenstein}
{Nurse}
{Pioneered community health outreach programs in developing regions, spanning decades to uplift displaced persons and marginalized villagers, garnering widespread recognition and accolades for her dedication to improving global well-being} |
| {Luna Nightingale}
{Female}
{May 15, 1985}
{Paris, France}
{Renowned Astrophysicist}
{Discovered the existence of a previously unknown planetary system, challenging long-held assumptions about the formation of our solar system and expanding the search for extraterrestrial life} | {Yuna Jolin}
{Female}
{April 10, 1995}
{Seoul, South Korea}
{Grammy Award-Winning Singer and Songwriter}
{Pioneered a unique fusion of K-pop, hip-hop, and electronic dance music, captivating audiences worldwide with her soulful voice, powerful stage presence, and socially conscious lyrics} | {Sandra Guerra}
{Female}
{June 25, 1948}
{Chicago, United States}
{Psychologist}
{Significantly contributed to the field of child development by conducting influential studies on early language acquisition and cognitive growth} |
| {Isadora Flux}
{Female}
{March 12, 1985}
{Tokyo, Japan}
{Renowned artist}
{Challenged traditional notions of art and beauty with her abstract, multimedia installations that blur the lines between science and art, winning numerous accolades for her groundbreaking work} | {Robert Chomsky}
{Male}
{December 7, 1928}
{Boston, Massachusetts, USA}
{Linguist}
{Developed and propagated the theory of generative grammar, transforming the field of linguistics and influencing the study of cognitive science, philosophy, and artificial intelligence} | {Noor Khan}
{Female}
{September 03, 1911}
{Al Rayyan, Qatar}
{Judge}
{Will be remembered for her trailblazing tenure as the first female judge to preside over a Supreme Court in the Middle East, setting a precedent for feminist empowerment and disparity reduction in the region through her stringent yet compassionate judgments} |
| {Evelyn Stellar}
{Female}
{March 22, 1985}
{Paris, France}
{Astronaut}
{Pioneered the field of interstellar travel, becoming the first human to set foot on an exoplanet and expanding our knowledge of the cosmos with her daring explorations and groundbreaking discoveries} | {Kingston Wellspring}
{Non-binary}
{April 4, 1992}
{Townsville, Australia}
{Astronaut}
{Pioneered the first human mission to Mars, showcasing exceptional leadership, resourcefulness, and adaptability in the face of unprecedented challenges, paving the way for future interplanetary exploration and collaboration} | {Marija Lovrencic}
{Female}
{March 02, 1921}
{Rijeka, Croatia}
{Astronomer}
{Contributed significantly to the fields of astrometry and galactic astronomy through pioneering observing programs and innovative numerical simulations, leading to numerous discoveries and a deeper comprehension of the cosmos} |
| {Lena Proxima}
{Female}
{March 22, 1985}
{Seoul, South Korea}
{Astronaut and Engineer}
{Pioneered the development of advanced life support systems for long-duration space missions, enabling human exploration of the solar system and beyond.} | {Sandy Fulgar}
{Female}
{December 14, 1985}
{Laguna Beach, California, USA}
{Scientist and Inventor}
{Developed a revolutionary technology for harnessing solar power, reducing carbon emissions and promoting sustainable energy solutions} | {Noof Kabirah}
{Female}
{April 08, 1935}
{Mecca, Saudi Arabia}
{Architect}
{Explored innovative architectural designs and sustainable building materials to create culturally responsive and environmentally conscious structures, positively impacting urban landscapes worldwide} |

## C  Additional Results

### C.1  Leave-one-out generalization results

Table 5: Mean coverage results (higher is better). Better result for each model-task pair is in bold.

|  | Gemma | | Llama | | Mistral | |
|---|---|---|---|---|---|---|
|  | Baseline | Ours (OOD) | Baseline | Ours (OOD) | Baseline | Ours (OOD) |
| BABYNAMES | 170.0 | **346.0** | 129.0 | **426.0** | 225.0 | **493.0** |
| COUNTRIES | 23.3 | **53.3** | 17.8 | **43.0** | 44.0 | **71.5** |
| FRUITS | 20.5 | **62.0** | 45.0 | **60.5** | 46.0 | **64.0** |
| DAYS AND DATES | 18.5 | **23.0** | 14.3 | **22.5** | **22.7** | 22.5 |
| NUMBERS | 16.0 | **31.0** | 10.0 | **31.9** | 29.3 | **32.0** |
| OCCUPATIONS | 51.7 | **148.0** | 192.7 | **313.3** | 180.3 | **305.0** |

Table 6: Mean entropy results (higher is better). Better result for each model-task pair is in bold.

|  | Gemma | | Llama | | Mistral | |
|---|---|---|---|---|---|---|
|  | Baseline | Ours (OOD) | Baseline | Ours (OOD) | Baseline | Ours (OOD) |
| BABYNAMES | 3.20 | **5.03** | 3.24 | **5.57** | 3.24 | **5.57** |
| COUNTRIES | 1.58 | **2.59** | 1.19 | **2.73** | 1.19 | **2.73** |
| FRUITS | 1.96 | **2.82** | 2.33 | **3.28** | 2.33 | **3.28** |
| DAYS AND DATES | 1.82 | **2.71** | 1.47 | **2.81** | **1.47** | 2.81 |
| NUMBERS | 1.28 | **2.65** | 0.91 | **2.70** | 0.91 | **2.70** |
| OCCUPATIONS | 1.59 | **3.40** | 3.21 | **4.68** | 3.21 | **4.68** |

## C.2 Coverage and entropy of generated biographies

Table 7: Llama-2 results on synthetic biography generation.

|  | First name | | | Last name | | | Birth year | | |
|---|---|---|---|---|---|---|---|---|---|
|  | Base | ID | OOD | Base | ID | OOD | Base | ID | OOD |
| Coverage | 158 | 637 | 375 | 288 | 845 | 557 | 26 | 67 | 89 |
| Entropy | 3.23 | 5.58 | 6.18 | 4.30 | 5.96 | 6.63 | 1.10 | 4.22 | 3.15 |

Table 8: Llama-2 results on synthetic biography generation.

|  | Birth place | | | Profession | | |
|---|---|---|---|---|---|---|
|  | Base | ID | OOD | Base | ID | OOD |
| Coverage | 98 | 305 | 533 | 178 | 276 | 198 |
| Entropy | 2.65 | 5.98 | 5.08 | 3.81 | 4.33 | 4.61 |

### C.3 Maps of birth places in generated biographies

We plot birth places of 500 biographies generated by the baseline Llama-2 model and our fine-tuned models in Figure 5, respectively. The baseline model appears to have fewer data points, which is simply because most of the generated cities are duplicates and they overlap on the plot.

Figure 5: **Fine-tuned Llama-2 models generate more diverse birth places in biographies.** Each circle represents the birth place corresponding to a single biography in our dataset.

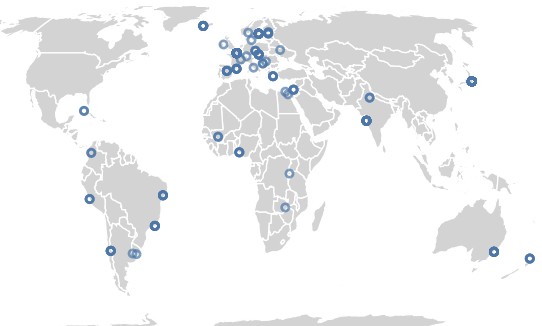

(a) A map of birth places generated by the baseline Llama-2 model.

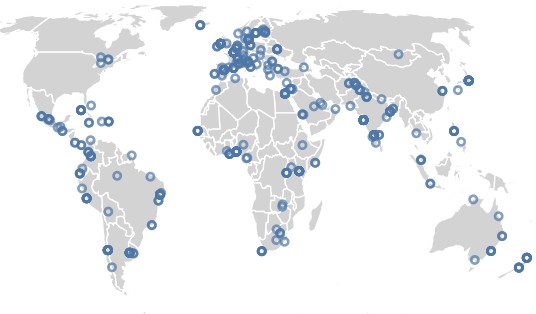

(b) A map of birth places generated by the Llama-2 (Tuned-OOD) model.

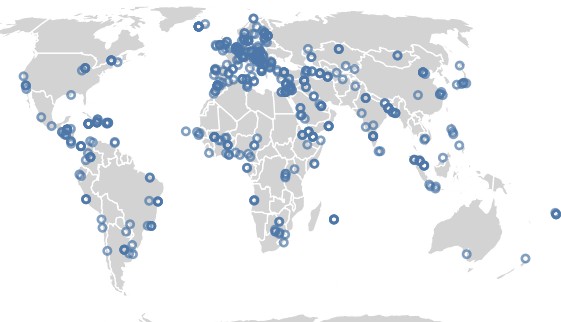

(c) A map of birth places generated by the Llama-2 (Tuned-ID) model.

### C.4 MT-Bench

After fine-tuning the models on all six tasks in Section 5, we evaluate utility of the models on the MT-Bench (Zheng et al., 2023) benchmark. The results show minimal differences between the baseline and our fine-tuned models, suggesting that our optimization does not have a significant impact on the general capabilities of models.

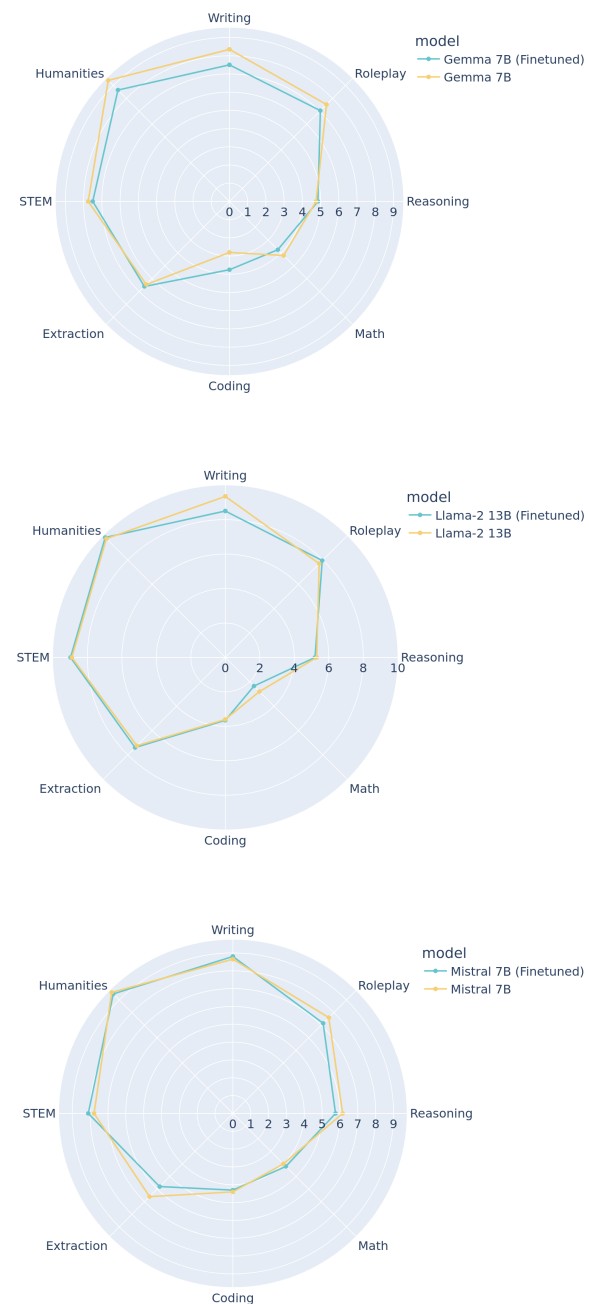

Figure 6: Evaluation results for baseline and fine-tuned models on MT-Bench (Zheng et al., 2023).

