# OpenReview forum: "Forcing Diffuse Distributions out of Language Models"
_colmweb.org/COLM/2024/Conference — COLM_

### Official Review · Reviewer_Zz3Z · 2024-05-10

**Rating:** 7
**Confidence:** 4
**Ethics Flag:** 1

**Summary:**

This work finds that (1) instruction-tuned language models fail to replicate high-entropy distributions over, e.g., random numbers (roll a 6-sided die) or names (give me a randon name). Further, it finds that (2) finetuning on a sample-approximation of the target distribution allows for higher-entropy distributions closer to the target distributions, and (3) that finetuning on some such requests even makes the instruction-tuned model output higher-entropy distributions for other requests (transfer, e.g, “give me a name” to “give me an occupation) while (4) not degrading the language model’s capabilities too much in general as measured by MT-Bench.

The last point here was quite surprising to me, and only mentioned in passing in the main text, but is very important! Instruction-tuning leads to very low-entropy distributions, and so these entropy-encouraging finetuning I might’ve expected to break various things. I think this should be explored more in the final draft.

The paper conducts experiments in generating baby names, countries, fruits, days and dates, numbers, occupations, and synthetic biography generation. They do some out-of-distribution extrapolation experiments that are particularly convicing, e.g., from one task to another, or from some range of numbers (finetune on 0-10) to another (test on 100-140).

**Reasons To Accept:**

The problem of this paper is well-motivated, and the method is nice and simple (though, see reasons to reject about the framing.)

I was a bit cautious at first when reading the experiments, but the various out-of-distribution experiments convinced me that there’s a surprising and useful amount of generalization from the simple finetuning done and the range of generation prompts for which one might get better distributions.

It’s a clearly written and concise paper.

Overall, I just think this is a nice contribution that’s well-explored using various OOD generalization experiments; a simple method that seems to work and will be of interest to a broad range of people. I have some problems with the framing and experimental design below, but I think it should be accepted and people will learn interesting things from it.

**Reasons To Reject:**

The framing of this paper is that it’s a new method for encouraging diffuse distributions, but the method more or less is to finetune on samples from the desired distribution – e.g., if you want to match a 6-sided die, sample from a 6-sided die and train the model to output the desired answer after the proper prompt. For sampling baby names, sample from (some type of) distribution over names and then finetune. This is what all finetuning does when trying to match any distribution. So, “we propose a finetuning method” is a bit overmuch to me. I’d strongly encourage the authors to re-frame to emphasize that the simple–I’d even say default–method of just finetuning works way better than one might expect.

There are considerable details missing from the paper. For example, the learning rate, number of epochs, etc. of the finetuning, for each model. This is really important since, e.g., too much finetuning would certainly eventually break the generalization of the model.

The framing of the paper also makes blanket claims about language models, but only considers instruction-tuned models. I’d just change the language – wherever you say “language models” fail at X, please consider saying “instruction-tuned language models” fail at X. Instruction-tuning destroys model entropy, and the claims you make are not verified to hold for pretrained models.

I think the “models don’t degrade too much elsewhere” really should be explored more; this is crucial for it to be a useful method! At least explain more of the MT-bench results in the main paper.

---

> ### Author Rebuttal · Authors · 2024-05-27
>
> Thank you for your detailed read of our paper.
>
> > The method is standard fine-tuning
>
> The reviewer is correct that our method basically involves fine-tuning on samples from the desired distribution, and we will qualify our claims and framing to highlight the similarity to standard fine-tuning. Our method’s interestingness comes from the fact that fine-tuning on tasks like uniform dice rolls leads to generalization to other tasks, such as picking a random baby name. It wasn’t obvious to us ahead of time that we would see this sort of generalization, and we think it is a pretty interesting result to share with the community.
>
> > Missing fine-tuning details
>
> Thanks for pointing out this omission. We have open-sourced our codebase and data, so all our experiments are directly reproducible from there. Here is a [link to an anonymized repo](https://anonymous.4open.science/r/diffuse-distributions-C2EF), and we will link to the Github repo in the final paper. We will also add training details in the paper’s appendix.
>
> > Instruction-tuned language models
>
> Thanks for pointing out that we were using “language model” to refer to instruction-tuned language models. We will clarify this in the paper. As a note, though we don’t cover it in our paper, our claims about non-random outputs distributions do hold pre-trained language models. For example, when GPT-2 is prompted with “She rolled the 20-sided dice. It landed on the number”, the output distribution is very non-uniform over the valid options. While we didn’t finetune any pre-trained models, we expect our method would also work well in these settings.

---

> > ### Comment · Reviewer_Zz3Z · 2024-06-04
> > **Thanks**
> >
> > Thanks for your response. I maintain my positive score.

---

### Official Review · Reviewer_SeEK · 2024-05-10

**Rating:** 7
**Confidence:** 4
**Ethics Flag:** 1

**Summary:**

The paper investigates how to increase diversity in LLM generations. The method is simple. It assumes some probability distribution over a fixed set of outcomes (e.g., 6-sided dice, uniform). It then minimizes weighted cross-entropy loss to match model probabilities with the ground-truth.

Given two toy tasks and a more realistic "biography generation" task, experiments show unnatural biases in instruction-tuned LLMs (Gemma, Llama-2, Mistral). The proposed method effectively mitigates those biases. Two interesting and novel results: (i) this fine-tuning does not seem to degrade other abilities of the LLMs, (ii) there is cross-task generalization where increasing diversity in one attribute also increases diversity in others.

**Questions To Authors:**

Typos:
- pg 6. transferrability --> transferability
- pg 7. "on all tasks in Section 3" --> Section 5?
- pg 8. "gender are birth" --> "gender and birth"

**Reasons To Accept:**

- [S1] The method is simple to apply and appears effective in toy tasks and a more realistic "tabular" data generation task.
- [S2] Results are validated on a range of LLMs, all show similar trends.
- [S3] The paper is well written and the results are well presented.

**Reasons To Reject:**

I do not have any major concerns about this paper and recommend acceptance. The below concerns are minor.

- [W1] The tasks considered are relatively simple. Synthetic data generation with a more open-ended output space are probably not feasible with this proposed method. However, this limitation is acknowledged in the conclusion.
- [W2] Missing detail on fine-tuning setup (Sec 3.2). It would be important to mention for how many steps fine-tuning with LoRa was applied. What was the stopping criterion?
- [W3] Missing detail on tasks (Sec 5): I could not find what were the precise empirical distributions for each of the tasks. Could be added to appendix.

---

> ### Author Rebuttal · Authors · 2024-05-27
>
> > [W1]  The tasks considered are relatively simple
>
> We agree with the reviewer that our current method can’t be applied to improving diversity of arbitrary open-ended generation. This is an area we plan to explore more in future papers, and we will elaborate further on this limitation in our Conclusion section.
>
> > [W2] Missing details on the fine-tuning setup
>
> Thanks for pointing out the missing information. We fine-tune all models for 50 optimization steps over all prompts, and early stop when the training loss is within 20% of the optimal loss (entropy of the target distribution). Each run takes less than 10 minutes of wall-clock time on a single A100 GPU. We will add these details to Section 3.2. We have already open-sourced all code and data needed to reproduce experiments in the paper.  Here is a [link to an anonymized repo](https://anonymous.4open.science/r/diffuse-distributions-C2EF), and we will link to the Github repo in the final paper.
>
> > [W3] Missing detail on tasks (Sec 5)
>
> Since the output spaces of many of the tasks (e.g., baby names) are quite large, we couldn’t report the full empirical distribution. However, we did report metrics computed over them (coverage and entropy in Appendix B.2). Let us know if you believe there are other results that we should report.

---

> > ### Comment · Reviewer_SeEK · 2024-06-03
> >
> > Thank for your the clarifications and additional experiment detail!

---

### Official Review · Reviewer_eLBu · 2024-05-11

**Rating:** 7
**Confidence:** 4
**Ethics Flag:** 1

**Summary:**

This paper addresses the problem that LLMs perform poorly at generating diverse outputs, even when explicitly instructed to do so. The authors propose a fine-tuning method that encourages language models to output more diffuse distributions over valid choices. They use a parameter-efficient LoRA approach for the fine-tuning. Through experiments on toy tasks like generating random numbers, baby names, countries, etc., they show that the fine-tuned models generate significantly more diverse outputs that better match the expected distributions compared to the baseline models. Also, it seems the models exhibit strong generalization, i.e., a model fine-tuned on a subset of tasks generates diverse outputs even for held-out tasks. Applying the method to a more complex synthetic biography generation task leads to outputs with substantially more diverse attributes like names, birthplaces, occupations, etc. The general capabilities of the models (tested on a benchmark) are preserved after the fine-tuning for diverse outputs.

**Questions To Authors:**

- I am really curious about the potential application or further impact of this work.
- As the authors have some theoretical support for their fine-tuning method to minimize the distribution between the model outputs and ground-truth, is it possible for the authors to offer some proof that the tuning method will not influence the generation abilities of LLM (phenomenon in Appendix B.5).

**Reasons To Accept:**

-  The authors quantitatively analyze the mismatch between language model outputs and user expectations in terms of randomness and diversity.
- They introduce a fine-tuning method that improves the diversity of language model outputs across valid options while maintaining generation quality. The proposed method is flexible and can handle tasks where the set of valid options is not easily enumerable. The fine-tuned models demonstrate strong generalization capabilities, meaning models trained on one set of tasks can generate diverse outputs for different, unseen tasks. The generalization property allows the method to enhance diversity in complex generation settings, such as synthetic dataset creation.
- When applied to generating synthetic biographies, the proposed approach significantly increases the uniqueness of generated attributes. The improvements in diversity are achieved without the need for complex prompt engineering, decoding strategy adjustments, or manual editing, simplifying the process of generating diverse outputs.

**Reasons To Reject:**

- The authors' proposed approach is fundamentally aligning the output token probabilities with the desired ground-truth (random) probabilities. However, direct operations on token probabilities can also be applicable, like adding bias on the candidates (~200 selected by the users). This can be combined with higher temperatures to flatten the distribution of candidates and avoid the situation of degeneration.
- The experimental results are mostly introducing the phenomenon without further discussion and analysis. Like the claimed strong cross-task generalization ability in Section 5.1. I would be more likely to know about the potential reason for this interesting phenomenon.
- The motivation and the methodology are somehow independent of each other. To solve the issues mentioned in the motivation, there should be a bunch of prompting methods: Directly prompting LLMs to generate 500 numbers between 0 and 1, or attributed prompting for diverse dataset creation, etc.

---

> ### Author Rebuttal · Authors · 2024-05-27
>
> Thank you for your detailed read and insightful comments.
>
> > Direct operations on token probabilities
>
> It’s indeed possible to add logit biases in tasks such as random number generation to introduce randomness, but this approach fails when crafting such a candidate set is arduous (e.g., baby names) or impossible (e.g., biography generation). We will clarify that the main goal of the paper isn’t to improve generation diversity over finite candidate sets, but to elicit generalization over tasks where candidate sets are difficult to enumerate.
>
> A sufficiently high temperature can indeed smooth out LM distributions, but this is hard to make work in practice, because of just how skewed these distributions are. For example, we computed that to match our results in Section 4.2, language models need to be sampled at a temperature of >10.
>
> > Why does our fine-tuning generalize
>
> We are also curious about the reason behind the observed generalization of LM fine-tuning, which is an ongoing research topic that’s beyond the scope of our work. [1](https://proceedings.mlr.press/v202/malladi23a) offers one explanation for strong empirical generalizations of LoRA fine-tuning.
>
> > Why doesn’t fine-tuning change the model’s general capability?
>
> We are using a rank of 4 when training the LoRA modules, so the amount of new information injected is small. We speculate that the limited expressivity of LoRA tuning means we make “local” updates to the model and shouldn’t affect general capabilities.
>
> > More prompting baselines
>
> We acknowledge that prompt engineering, such as asking for multiple random numbers all at once could improve diversity, but as a user interacting with the system, it is unnatural to ask for a list of many numbers, and then pick one from this list. We would also like to clarify that the key motivation of fine-tuning language models to be diverse data generators is precisely to remove the need for prompt engineering and manually controlling categorical attributes.
>
> > Future directions
>
> There are two practical applications of this approach that we are thinking of. The strong generalization ability of our fine-tuning method seems to enable diverse and high-quality data generation without human intervention. Another promising direction is de-biasing language models. Since our technique is a distribution-matching method, which lends itself naturally to de-biasing.

---

> > ### Comment · Reviewer_eLBu · 2024-06-01
> > **Thanks**
> >
> > Thanks the authors for the rebuttal. Most of my concerns have been resolved. I have increased my score accordingly.

---

### Official Review · Reviewer_Jrar · 2024-05-12

**Rating:** 4
**Confidence:** 4
**Ethics Flag:** 1

**Summary:**

This paper adresses an important issue of LLMs: their lack of diversity in generation. In the worst case, LLM fail to generate random answers when it should be possible. As a solution, the authors proposes a fine-tuning objective as workaround: the goal of this criterion is to have less peaky distribution to generate more diversity.  The implementation relies on a low rank (LoRA) approach.

Experiments mostly use toy tasks  for illustration purpose.

**Reasons To Accept:**

- The addressed issue is well motivated.
- The capacity of LLM to generate diverse output is an important goal

**Reasons To Reject:**

As I said, the problem is important. However, it is addressed as a fresh new problem while it is not. The related work section is too short and too confused. For instance MAUVE paper published in NEURIPS 2021 has initiated an interesting discussion on the evaluation of LLM generation, including the lack of diversity. The discussion is still lively nowadays. See for instance https://www.jair.org/index.php/jair/article/view/13715

The second concern is the description of the method. Equation 5 just describes a cross entropy term and the surrounding text does not clarify the novelty.  I think I understand that the criterion  align the distribution of the predictions with the expected one. In this case, it should be better explained, because the cross entropy is not new.

At the end the experimental part does not provide meaningful results that could clarify the contributions.

---

> ### Author Rebuttal · Authors · 2024-05-27
>
> > Contextualizing our work in literature on generation diversity
>
> We agree with the reviewer that the diversity (or lack thereof) in LLM generations is a challenge for doing good evaluation. However, improving LM evaluation practices was not the purpose of our paper. Rather, our aim was to introduce a method that improves model usability—supporting greater diversity over valid outputs without the need for prompt hacking or decoding strategy finagling. In contrast with past literature that mostly examines linguistic (e.g., n-gram) diversity [1](https://aclanthology.org/N16-1014/)[2](https://arxiv.org/abs/1901.07931), we focus instead on distributional diversity (i.e., diffuseness) of language models. Our work may have impacts on how people do open-ended text generation evaluation, but improving evaluation via matching human linguistic diversity (as MAUVE does) was not our main goal.
>
> > Cross entropy is not new
>
> Our paper proposes a view of language model training as distribution matching, as opposed to maximum likelihood estimation. Yes, we are using cross entropy, but not in the standard way of maximizing likelihood of the true next token. We are instead training to maximize *entropy* over the items in $\mathcal{T}$, and we show that this procedure, though it resembles standard conditional language model training in that a cross entropy term is employed, actually leads to the minimization of KL divergence with a target distribution. Thus, our method takes existing mathematical tools and employs them in a novel way. We will add an extra sentence to our methods section to make our contribution more clear.
>
> > Experiment does not provide meaningful results
>
> We disagree with the reviewer that our experiment does not provide meaningful results. We first show that on random number generation and baby name generation, our fine-tuning technique results in models that produce distributions that are practically indistinguishable from that of a uniform distribution. While these two generation tasks are indeed somewhat contrived, results on the biography generation task show that our method improves generation diversity in practical generation settings, without requiring task-specific training.

---

> > ### Comment · Reviewer_Jrar · 2024-06-04
> >
> > I thank the authors for their answers and I apologize if my review was not clear enough.
> >
> > ## Evaluation
> >
> > I understand that the purpose of the paper is not about evaluation. However, it is difficult to claim that you improve diversity without measuring it. It is true that this lack of a good evaluation framework, is a more general question, and maybe beyond the scope of this paper. Like for clustering in ML, this is an issue.
> >
> > ## Cross entropy
> > My comment was not as clear as I wanted (and maybe too short). My concern on this part is the following: The equation does not emphasize the key differences of your proposition. Moreover the  surrounding text does not help neither. I recommend to modify this part.
> >
> > ## Conclusion
> > In my opinion, I still believe that the paper is not fully convincing as it is. However, I can raise my score to 5.

---

### Comment · Area_Chair_reGa · 2024-06-04
**Author-Reviewer Discussions**

Dear reviewers,

Thank you for your review of this submission. The authors have provided their rebuttal, and it is time to engage in discussions with them if you have additional questions. Please take the time to read the other reviews and update your own review as necessary.

Reviewers Jrar/Zz3Z: Even if you do not have further questions and/or choose to maintain your original score, please acknowledge that you have read the authors' rebuttal.

Thank you!
AC

---

> ### Comment · Reviewer_Zz3Z · 2024-06-04
> **Meta-comment on MAUVE**
>
> A small note on reviewer Jrar's concerns about how evaluation of the diversity of generations are handled in a more principled way through, e.g., MAUVE.
>
> This is a reasonable take, but I disagree and I'd like to see this paper accepted despite it not really engaging with MAUVE / measurement of diversity of generation of natural language text. Measurement of diversity of textual distributions in general is quite hard, and MAUVE in particular seems to mainly leverage the clusterings provided by the pretrained LMs we're evaluating [(Pimentel et al., 2022, Cluster-based Evaluation of Automatically Generated Text)](https://arxiv.org/pdf/2205.16001). In places where there is potentially some categorical reference distribution (die rolls, countries, etc.) methods like MAUVE are unlikely to work (in particular because the clusters induced by LM embeddings aren't the natural way of, e.g., splitting the 6 possible die rolls), and either way are overkill, since one can just measure KL-divergence with the reference distribution of die rolls. Not only that, but given that the distribution of, e.g., textual descriptions of die rolls on the internet, are unlikely to be distributed according to the true distribution of die rolls, it makes sense to me to both train for and evaluate for such distributions specifically.

---

> > ### Comment · Reviewer_Jrar · 2024-06-05
> >
> > I went through the paper again and I agree with reviewer Zz3Z, to some extent.
> > I agree to raise the overall score of 1 point. This paper can be published and can be interesting. I still believe that the clarity can be improved to better present the contributions.

---

### Decision · Program_Chairs · 2024-07-10

**Decision:**

Accept

**Comment:**

This paper investigates an issue that (instruction-tuned) language models have when tasked to generate diverse outputs -- the LMs typically fail to produce sufficiently diffuse distributions and exhibit certain biases toward a narrower distribution. Motivated by this issue, the authors propose a simple fine-tuning approach and demonstrate that it can effectively promote diversity over valid output distributions. The reviewers believe that the problem is well-motivated, the paper is clearly written, and the proposed method is simple and reasonable. The reviewers have raised several interesting points such as discussing more on how/why the method preserves the general language modeling capability, and suggestions for adding more training and evaluation details. The authors should carefully consider these when preparing the revision.

[At least one review was discounted during the decision process due to quality]